# Analytical protocol for measuring micro-molar quantities of sulfur volatile species in experimental high pressure and temperature fluids
Arianna Secchiari [1] ✉, Luca Toffolo [1], Sandro Recchia [2] & Simone Tumiati [1]

Validating thermodynamic models is essential in experimental geosciences for exploring increasingly complex systems and developing analytical protocols. However, investigating solid–fluid equilibria in mm$^3$-sized experimental capsules poses several challenges, particularly in sulfur-bearing chemical systems. These include maintaining bulk fluid composition and performing quantitative analysis with extremely low amounts of synthesized fluid. We present an innovative methodology for measuring ultra-low amounts of sulfur volatiles ($H_2S$ and $SO_2$) generated during experimental runs at high pressure and temperature conditions of 3 GPa and 700 °C. Using solid sulfides ($FeS + FeS_2$) and water as reactants, we performed redox-controlled syntheses employing a piston cylinder apparatus. We demonstrate that ex-situ measurements of these fluids by quadrupole mass spectrometry ensure accurate and precise analysis, confirming predicted thermodynamic compositions. This methodology allows in-depht investigation of sulfide solid–fluid equilibria, shedding light on sulfur volatiles behavior and geochemical cycles under high $P$–$T$ conditions characteristic of the Earth's interior.

The cycling of volatile elements, such as carbon and sulfur, within the Earth plays a pivotal role in shaping the long-term composition of the terrestrial reservoirs. These processes significantly influence our planet's habitability, climate regulation, and potentially even the evolution of life[1].

Subduction zones are regions on Earth where one tectonic plate (subduction slab) is pushed under another for hundreds of kilometers, leading to significant geological activity, such as massive release of volatiles, earthquakes, and volcanic activity at shallow depths with the formation of new crust. These areas also serve as crucial links in geochemical cycles, facilitating the transfer of volatile compounds from the surface (atmosphere, hydrosphere, and biosphere) into the deep Earth's interior[1], which eventually go back into the atmosphere through arc volcanism and associated gaseous emissions[1,2]. At the slab-mantle interface, which occurs around 100–130 km depth below continental arcs, where high pressure and temperature conditions (HP–HT; ~3–4 GPa and 700–900 °C) prevail, chemical exchanges among the different terrestrial reservoirs are primarily governed by volatiles-driven reactions due to dehydration of hydrous rock-forming minerals[3]. However, due to the inaccessibility of the Earth's inner reservoirs, these processes remain largely unconstrained.

Volatile elements combine in the Earth's interior, forming "fluids", which are low-density and low-solute mixtures of water and non-polar species, typically $CO_2$ and $CH_4$ (C–O–H or COH fluids). Since pioneering experimental petrology studies arose back in the sixties[4], significant progress in understanding solid–COH fluid equilibria within the Earth's interior has been achieved experimentally by replicating geologically meaningful HP–HT conditions in the laboratory. This is accomplished by conducting fluid synthesis within low-volume (<20 mm$^3$) inert noble metal capsules using an end-loaded piston cylinder apparatus, followed by ex-situ analyses of the produced fluids[5]. While this methodology allows for fluid synthesis across a wide range of pressure-temperature ($P$–$T$) conditions and enables control of oxygen fugacity through the double capsule strategy[4], a significant limitation arises from the small volume of fluid produced. This low fluid amount presents an analytical challenge in achieving accurate and high-precision measurements, along with the critical task of preserving the bulk composition of the fluids synthesized at HP–HT conditions upon quenching them to room temperature and pressure. As for COH-fluid bearing systems, ex-situ analyses conducted on fluids equilibrated in experimental capsules and quenched to room temperature and pressure

[1]Dipartimento di Scienze Della Terra, Università Degli Studi di Milano, via Mangiagalli 34, I-20133 Milano, Italy. [2]Dipartimento di Scienza e Alta Tecnologia, Università degli Studi dell'Insubria, via Valleggio 11, I-22100 Como, Italy. ✉e-mail: arianna.secchiari@unimi.it

conditions were proven to be among the most effective techniques for quantitatively determining volatile components[6,7]. According to these procedures, after the experiment, the fluids are extracted and analyzed by gas-chromatography or mass-spectrometry, guaranteeing high analytical performance. However, in contrast to the extensive literature available for carbon-bearing systems[7–9], the synthesis and comprehensive characterization of S-bearing (i.e., SOH) aqueous fluids in the laboratory remains an analytical challenge.

Among the volatile species, sulfur (S) plays a pivotal role in numerous geochemical reactions occurring in a wide range of pressures and temperatures. Its ability to exhibit multiple oxidation states, with sulfide ($S^{2-}$) and sulfate ($SO_4^{2-}$) being the most common forms, makes sulfur particularly intriguing due to its sensitivity to the diverse redox states of various systems. For this reason, the experimental investigation of sulfur behavior and speciation in geological fluids at different $P–T$ and redox conditions has drawn significant attention[10–13]. Nevertheless, the primary difficulty in dealing with S-bearing experimental systems lies in sulfur's high reactivity, which hinders the preservation of the original chemical composition of the generated fluids. As a result, accurate measurements of such fluids, particularly $H_2S$, have been impeded by the absence of robust chemical protocols specifically designed for this purpose. Until now, the presence of $H_2S$ in experimental fluids has only been identified qualitatively by the characteristic "rotten egg" smell[13].

To bridge this knowledge gap, we have performed experiments to generate SOH fluids under HP–HT ($P = 3$ GPa and $T = 700$ °C) and controlled redox conditions, using solid sulfides and water as starting materials. Here, we propose an analytical protocol specifically designed for the synthesis and measurements of sulfur-bearing volatile species. We will demonstrate that this new methodology validates the existing thermodynamic models. By enabling fast, accurate and high-precision measurements of sulfur-bearing volatile compounds in ultra-low fluid amounts, our protocol can be regarded as a new reference for the characterization of SOH fluids. This method lays the groundwork for a more comprehensive investigation of SOH fluids in more chemically complex systems, useful as analogical models of geological environments but also for technological applications, allowing for the assessment of chemical equilibria and state relationships in increasingly complex thermodynamic systems.

## Results and discussion

Three syntheses of redox-buffered SOH fluids were conducted at 3 GPa and 700 °C (Table 1) using the double capsule technique (Fig. 1). To validate our methodology, we employed iron–wüstite (IW) and fayalite–magnetite–quartz (FMQ) redox buffers (Fig. 2a, b) to explore a range of potential fluid compositions under relatively reducing and oxidizing conditions. According to thermodynamic models, the IW-buffered experiment is predicted to produce a fluid with low $H_2S$ content along with the pyrrhotite formation at the expense of pyrite, as highlighted by Eq. (1):

$$FeS_2 \text{(pyrite)} + H_2 \rightleftharpoons FeS \text{(pyrrhotite)} + H_2S \qquad (1)$$

In contrast, the FMQ-buffered run should result in a $H_2S$-free fluid composed entirely of $H_2O$, thus serving as our blank control. For this experiment, as $H_2S$ formation—and therefore pyrrhotite generation—is not predicted, the initial pyrite/pyrrhotite ratio is expected to remain unchanged.

For our experiments, a runtime of 5-h was set for SOH-IW2 and SOH-FMQ1 runs, while a 6-h runtime was chosen for run SOH-IW1 (Table 1). This duration was based on previous experimental works conducted on magmatic systems[14]. Since these studies were performed at higher temperatures ($T = 1300$ °C) compared to our experiments, we initially set longer runtimes ranging from 1 week to 8 to 10 h. Additional details, including a complete list of the experiments conducted, are reported in Supplementary Tables 1 and 2. However, in all the IW-buffered experiments exceeding 6 h,

**Table 1 | Volatile speciation of the SOH fluids synthesized under controlled $P–T$–redox conditions and measured by quadrupole mass spectrometry**

| Synthesis | SOH-IW1 | SOH-IW2 | SOH-FMQ1 |
|---|---|---|---|
| $P$ (GPa) | 3 | 3 | 3 |
| $T$ (°C) | 700 | 700 | 700 |
| Buffer assemblage | IW | IW | FMQ |
| Runtime (h) | 6 | 5 | 5 |
| Fluid phase | | | |
| µmol tot | 36.05 | 106.72 | 31.68 |
| µmol | | | |
| $H_2O$ | 34.76 (0.08) | 78.53 (0.14) | 31.68 (0.10) |
| $H_2$ | bdl | 9.47 (0.02) | bdl |
| $H_2S$ | 1.03 (0.07) | 14.06 (0.07) | bdl |
| $SO_2$ | bdl | 4.66 (0.09) | bdl |
| mol % | | | |
| $H_2O$ | 97.1 (0.22) | 73.6 (0.13) | 100.0 (0.32) |
| $H_2$ | – | 9.00 (0.02) | – |
| $H_2S$ | 2.9 (0.20) | 13.2 (0.07) | – |
| $SO_2$ | – | 4.36 (0.08) | – |
| Thermodynamic model | | | |
| $H_2O$ | 86.65 | 86.56 | 99.99 |
| $H_2$ | 1.2 | 1.2 | – |
| $H_2S$ | 16.15 | 16.15 | 0.01 |
| $SO_2$ | – | – | – |

The total amount of fluid synthesized is expressed in µmol and calculated from the ideal gas law $PV = nRT$. The amount of the monitored species (µmol) derived from linear regression analysis performed through a specifically designed Wolfram Mathematica® routine. The volatile speciation of the SOH fluid is expressed as moles percentage on an air- and $N_2$-free basis (mol% *). The thermodynamic model is reported for comparison. See the main text for further details. *bdl* below detection limit.

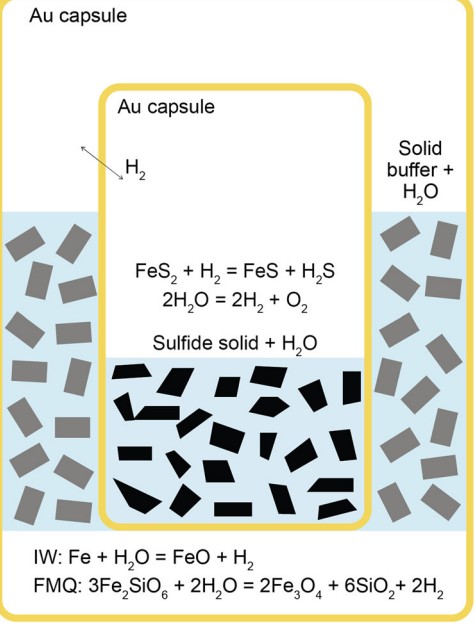

**Fig. 1 | Schematic representation of the double Au capsule setup used to buffer $fH_2$ in our experiments.** The relevant $fH_2$-buffering reactions are shown together with the equilibria that involve the SOH fluid in the inner capsule.

**Fig. 2 | Electron microscope images showing the microtextures of the buffer assemblages and run products for experiments SOH-IW2 and SOH-FMQ1.** Buffer assemblages: **a** iron–wüstite (i.e., Iron–Wü) in SOH-IW2 and **b** ferrosilite–magnetite–quartz (i.e., Fs–Mag–Q) in SOH–FMQ1. Run products: **c** newly formed pyrrhotite (Po) derived from pyrite (Pyr) reaction in experiment SOH-IW2; **d** pyrite (Pyr) crystals in experiment SOH-FMQ1.

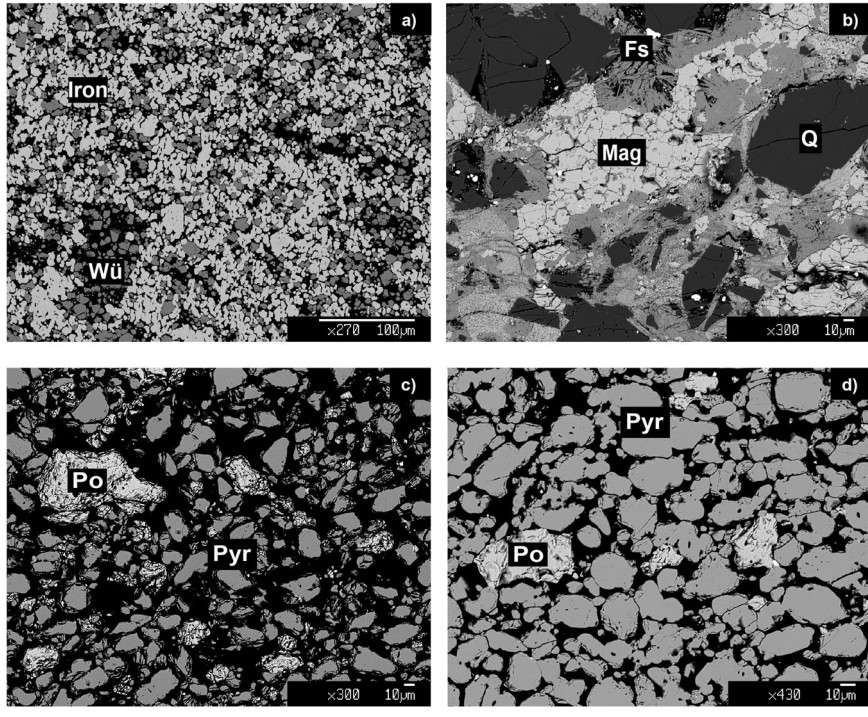

the inner capsule was invariably found devoid of fluids (Supplementary Table 1), suggesting that runtimes beyond 5 h may trigger capsule weakening due to the produced $H_2S$, resulting in fluid loss during quenching.

In all the experiments conducted under reducing conditions, i.e., SOH-IW1 and SOH-IW2, microtextures revealed the consumption of pyrite to form pyrrhotite as the solid run product (Fig. 2c), in agreement with thermodynamic predictions. In contrast, in the FMQ-buffered experiment (SOH-FMQ1), the initial pyrite/pyrrhotite ratio appears to be preserved (Fig. 2d).

Image analysis performed on X-ray compositional maps (Fig. 3, Supplementary Fig. 1) clearly illustrates the growth of pyrrhotite at the expense of pyrite (Fig. 3a–c) especially towards the inner-capsule walls (Supplementary Table 3). Remarkably, no reaction between the produced sulfur-bearing volatile species and either the inner or outer capsule was evidenced (Fig. 3b). This suggests that fluid loss did not occur during the experiment, confirming that Au is a suitable material for this type of synthesis.

X-ray compositional maps also indicate that the width of the pyrrhotite-rich layer is more pronounced in the 6-h run (SOH-IW1, Fig. 3d–f and Supplementary Table 3) compared to the 5-h run (SOH-IW2, see Fig. 3a–c). This observation provides compelling evidence that the pyrrhotite-forming reaction reported in Eq. (1) has occurred, driven by $H_2$ diffusion from the outer Au capsule, progressing from the edge of the inner capsule toward its central part.

In contrast, the FMQ-buffered experiment SOH-FMQ1 confirms that the initial pyrite/pyrrhotite ratio was preserved, with no precipitation of newly formed pyrrhotite (Fig. 3g–i and Supplementary Table 3).

Our results underscore the critical importance of runtime in determining capsule resistance and the successful outcome of the experiment. This is evident from runs SOH-IW1 and SOH-IW2, conducted under identical $P$, $T$, and redox conditions but with slightly different runtimes (6 and 5 h, respectively; see Table 1). The fluid synthesized in the experiment SOH-IW1 (Table 1) generated a $\Delta P$ of 39 mbar in the piercing chamber, which, according to the volume and the temperature of the chamber itself, corresponds to 36.05 µmol of total volatiles. The fluid primarily consisted of $H_2O$ (97.1 mol%), with minor amounts of $H_2S$ (2.9 mol%, Supplementary Fig. 2). In contrast, the fluid released from the experiment SOH-IW2 generated a $\Delta P$ of 128 mbar, corresponding to 106.72 µmol of volatiles. The analyzed fluid (Fig. 4a–e) is composed of $H_2O$ (73.6 mol%), $H_2S$ (13.2 mol%), $H_2$ (9.0 mol%)

and $SO_2$ (4.36 mol%). This measured bulk fluid composition is nearly identical to that predicted by thermodynamic models for chemical equilibrium (Fig. 4a). However, the chemical speciation of the analyzed fluid deviates from thermodynamic predictions (Table 1), showing higher $H_2$ and lower $H_2O$ contents, even with some oxidized sulfur ($SO_2$) despite of the highly reducing conditions. This apparent discrepancy may be explained by a back-reaction likely occurring in the fluid during the temperature drop imposed at the end of the experiment, which changed the pristine speciation of the fluid at high P-T conditions. Hydrogen and $SO_2$ could have formed from water and $H_2S$, a reaction that is known to occur spontaneously at hydrothermal conditions[15], as described by Eq. (2):

$$H_2S + 2H_2O \rightleftharpoons 3H_2 + SO_2 \qquad (2)$$

The preservation of the bulk fluid composition in experiment SOH-IW2 strongly suggests that these changes in fluid speciation are merely the result of a partial late-stage chemical re-equilibration during quenching to room conditions. These results align with previous findings on COH systems, which have demonstrated that ex-situ techniques effectively preserve the bulk fluid composition[8,9] but not necessarily the fluid speciation[7,8], which is highly sensitive to pressure and temperature conditions and may be altered during quenching. In this framework, the development of rapid quenching techniques will be of crucial importance for the investigation of natural systems and, in particular, of geological fluids. While still in the early stages of implementation[16], these methods utilize faster cooling rates compared to those routinely employed in experimental petrology laboratories. This advancement holds the potential to provide more accurate representations of fluid speciation under HP–HT conditions, thereby enhancing our understanding of fluid behavior in geological processes.

The results obtained from experiment SOH-IW2 confirm that chemical equilibrium between the solid and fluid phases is achieved within five hours. The lower $H_2S$ content in experiment SOH-IW1, which ran for six hours, compared to experiment SOH-IW2, appears counterintuitive to chemical kinetics principles. This behavior is due to $H_2S$ loss due to diffusion, as evidenced by a strong "rotten egg" smell when the outer capsule was peeled off. As previously stated, no reaction with the inner or outer capsule was observed during microprobe analysis

**Fig. 3 | Back-scattered electron images of solid run products and compositional X-ray maps of sulfur and iron for the performed experiments.** Solid run products (i.e pyrrhotite = Po and pyrite = Pyr) of the different runs are reported as follows: **a** SOH-IW2; **d** SOH-IW1; **g** SOH-FMQ1. Compositional X-ray maps of sulfur for the performed experiments are illustrated in **b** SOH-IW2; **e** SOH-IW1; **h** SOH-FMQ1. Compositional X-ray maps of iron are shown in **c** SOH-IW2; **f** SOH-IW1; **i** SOH-FMQ1. Compositional X-ray maps scale is expressed in count per second for both sulfur and iron. Image analysis was performed through a specifically designed Wolfram Mathematica® routine.

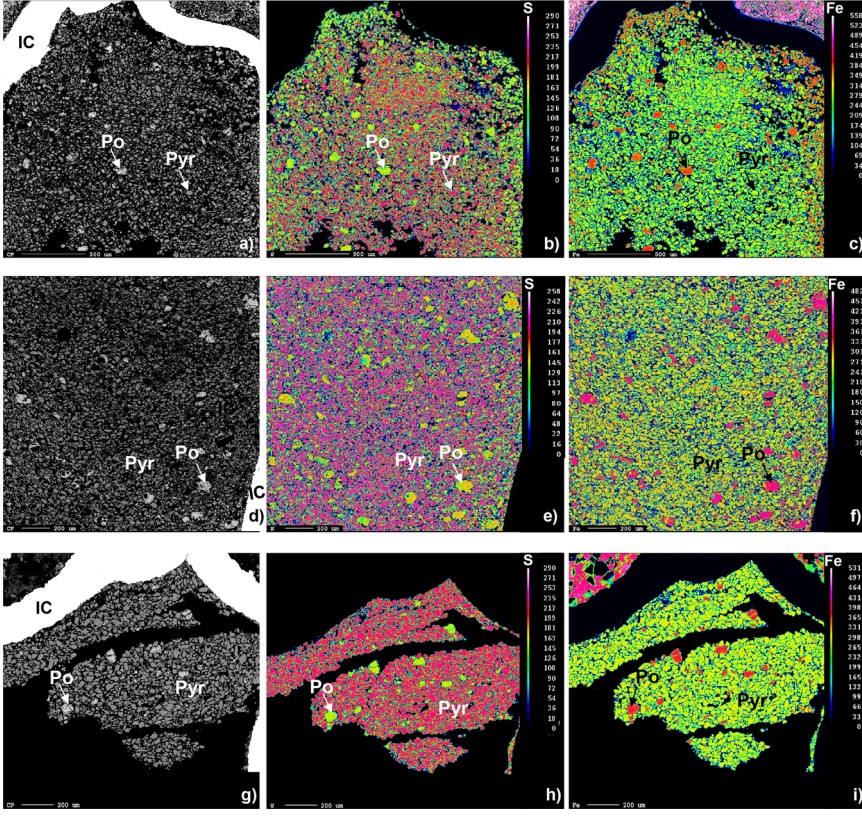

(Fig. 3b, e), indicating that fluid loss occurred at the end of the experiment, i.e., during quenching. Nevertheless, experiment SOH-IW1 is crucial for developing our protocol for several reasons. First, the extremely low $H_2S$ contents (1.03 µmol ± 0.072, see Table 1 and Supplementary Fig. 2) obtained in experiment SOH-IW1 provide an excellent opportunity to rigorously test the analytical sensitivity of the QMS measurements. Our data unequivocally show that, despite the low amount of produced $H_2S$, the proposed protocol ensures high-precision measurements of sulfur volatiles. This is evidenced by the extremely low standard deviation values (7 mol%) on $H_2S$ measurements.

Second, this experiment demonstrates that once chemical equilibrium is achieved, the produced $H_2S$ begins to weaken the inner capsule, leading to fluid diffusion during quenching. Therefore, experiments should not exceed 5 h, i.e., the time required to reach chemical equilibrium.

A series of additional experiments (Supplementary Results), conducted under different pressure and temperature conditions (Supplementary Table 2, Supplementary Figs. 3 and 4), shows that temperature is a key factor in governing the kinetics of the $H_2S$-forming reaction, while pressure has minimal impact. These preliminary findings suggest that our protocol, although optimized for 3 GPa and 700 °C, can be effectively adapted across a broader range of pressure and temperature conditions.

Based on the results obtained from experiment SOH-IW2, we ran our blank control, experiment SOH-FMQ1, for five hours. The produced fluid (Table 1 and Fig. 4f) consists of pure $H_2O$ (100 mol%, Fig. 4g–l) and is generated at $\Delta P$ of 38 mbar. As for experiment SOH-IW2, the fluid bulk composition overlaps with that predicted by thermodynamic models (Fig. 4f).

The consistency between the experimental data and thermodynamic models suggests that the bulk composition in the fluid was preserved throughout quenching and fluid measurement. The proposed protocol thus represents a robust and reliable methodology, optimized for specific pressure and temperature conditions for quantitative analysis of ultra-low amounts of sulfur-bearing fluid.

## Conclusions

Our study pioneers a robust and reliable protocol for synthesizing and analyzing ultra-low amounts of sulfur-bearing volatile species generated under high pressure and temperature conditions of 3 GPa and 700 °C and controlled redox state. By rigorously maintaining the bulk fluid composition validated against thermodynamic models for chemical equilibrium, the proposed methodology ensures fast and high-precision analysis of the sulfur volatile species in the produced fluid.

This protocol sets a new benchmark in the experimental investigation of sulfur-bearing chemical systems, opening avenues for more in-depth studies of sulfur species behavior in complex geological environments, particularly in subduction settings. Future perspectives will involve understanding sulfur behavior and identifying which species are dominant at fixed *P–T* and redox conditions. The adaptability of the methodology across varying *P–T* conditions also paves the way for potential work over a broad range of *P–T* conditions. Investigating the interaction of sulfur species with redox-sensitive elements, such as transition metals, under varied *P–T* conditions will provide crucial insights into their distribution, stability and cycling in subduction settings, with important economic implications. Additionally, examining sulfur behavior in response to redox changes could reveal sulfur's broader implications in deep Earth processes, contributing to a more comprehensive understanding of volatile cycles and magma oxidation in subduction zones.

## Methods

### Experimental approach and synthesis of the SOH fluid

In this study, SOH fluids were synthesized from a sulfide powder consisting of a $FeS_2$ (pyrite) and FeS (pyrrhotite) (Sigma-Aldrich) mixture (9:1 molar ratio) reacting with water at redox-buffered conditions. Experiments were buffered using the so-called "double capsule technique"[4] to prevent direct contact between the sample and the buffering assemblages. The setup included an inner, Au capsule (outer diameter OD = 2.3 mm, inner diameter ID = 2.1 mm), containing ~40 mg of sulfide powder and ~2 µl of ultra-pure MilliQ® water, and an outer Au capsule (OD = 4.5 mm, ID =

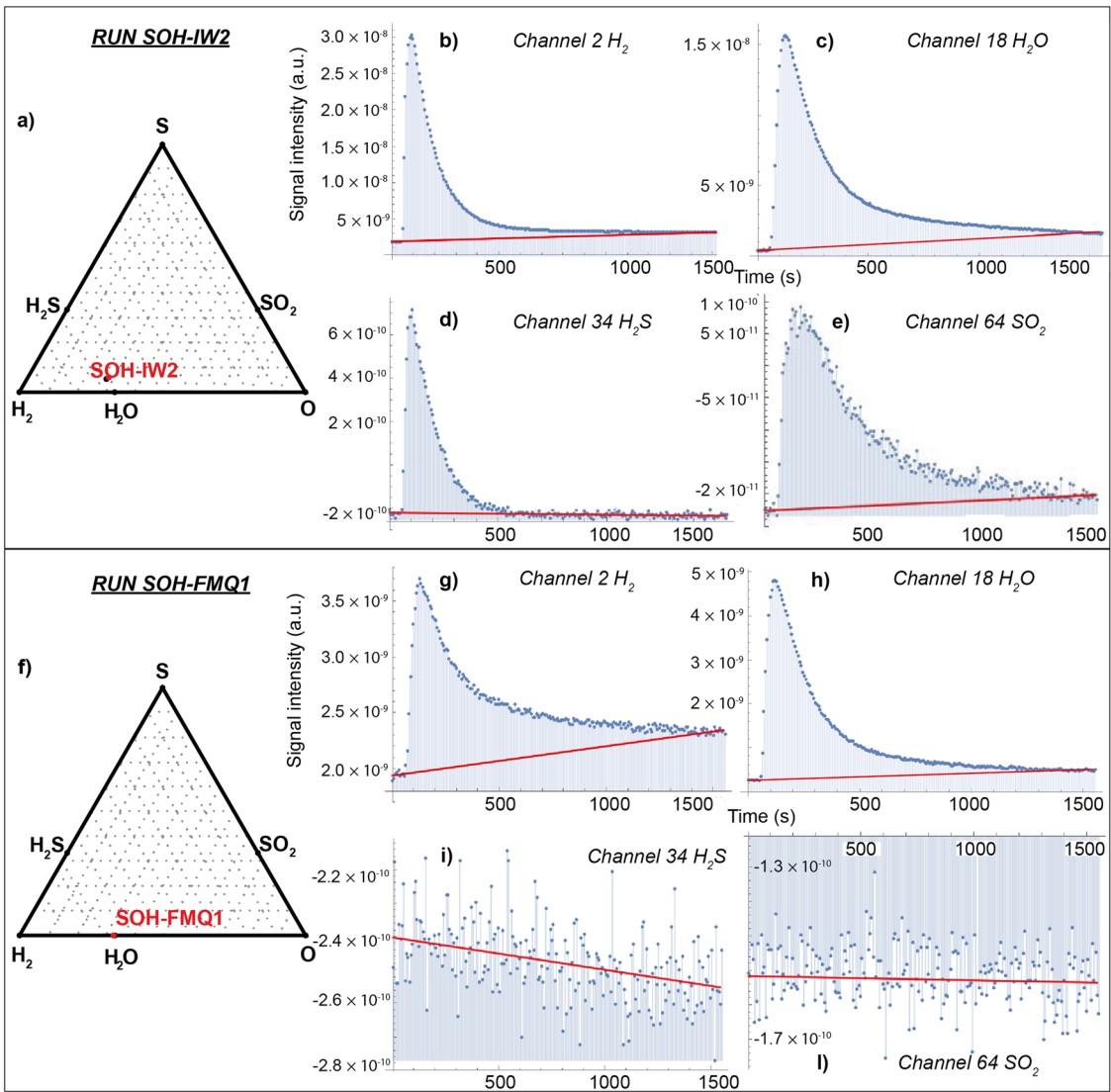

**Fig. 4 | SOH ternary diagram for sulfide-saturated SOH fluids buffered at $f$H$_2$$^{IW}$ (run SOH-IW2) and $f$H$_2$$^{FMQ}$ (run SOH-FMQ1) and related $m/z$ signal integration (peak area) for the measured volatile species.** Run SOH-IW2: **a** bulk fluid composition measured by the capsule-piercing QMS (red dot) compared to fluid composition predicted by the thermodynamic model using Eos[18] (black dot). The size of the red dot includes the analytical uncertainty. Signal integration (peak area) expressed as $m/z$ for the different channels (i.e., measured volatile components) measured as partial pressures over time: **b** channel 2: H$_2$; **c** channel 18: H$_2$O;

**d** channel 34: H$_2$S; **e** channel 64: SO$_2$. The red line represents the background signal. Run SOH-FMQ1: **f** bulk measured fluid composition (red dot) compared to the predicted composition (black dot). The size of the red dot includes the analytical uncertainty. Signal integration (peak area) expressed as $m/z$ for the different investigated channels: **g** channel 2: H$_2$; **h** channel 18: H$_2$O; **i** channel 34: H$_2$S; **l** channel 64: SO$_2$. For both experiments quantitative analysis was performed by integration of the peak area through a specifically designed Wolfram Mathematica® routine.

4.1 mm), filled with a redox buffering assemblage soaked in water (Fig. 1). It is noteworthy that, normally, an alloy containing Pd and Pt is used for the inner capsule with the double capsule technique to ensure better hydrogen permeability. However, Pt and Pd are not recommended when dealing with S-bearing fluids containing H$_2$S due to their tendency to react by forming sulfides, thereby altering the composition of the investigated system and the mechanical properties of the capsules themselves. Therefore, we decided to use thin, 0.1-mm wall inner capsules made of gold. We have verified, by replicating experiments in the COH system and comparing our findings to the thermodynamic model, as well as with previously published experimental data[9], that under our experimental conditions, inner gold capsules are also permeable to hydrogen. Details and results of this test experiment are reported in Supplementary Methods.

As for the redox-buffering assemblages, we used: (i) iron–wüstite (IW), to impose relatively reducing conditions in the inner capsule; and (ii)

fayalite–magnetite–quartz (FMQ; actually forming ferrosilite–magnetite–coesite at run conditions), to impose relatively oxidizing conditions. As long as all the buffer phases are present, which has been subsequently verified through scanning electron microscopy and electron microprobe analysis (Fig. 2a, b), the hydrogen chemical potential must be homogeneous in the inner and outer capsule. Further details on microprobe analysis are provided in Supplementary Methods. In the outer capsule, the hydrogen fugacity ($f$H$_2$) is constrained through Eq. (3) for the IW buffer and Eq. (4) for the FMQ buffer:

$$\text{IW}: \text{Fe} + \text{H}_2\text{O} \rightleftharpoons \text{FeO} + \text{H}_2 \qquad [\log(f\text{H2}/1\text{bar}) = 5.62] \qquad (3)$$

$$\text{FMQ}: 3\text{Fe}_2\text{Si}_2\text{O}_6 + 2\text{H}_2\text{O} \rightleftharpoons 2\text{Fe}_3\text{O}_4 + 6\text{SiO}_2 + 2\text{H}_2 \qquad [\log(f\text{H2}/1\text{bar}) = 2.20]$$
$$(4)$$

In the inner capsule, the equilibration of the SOH fluid is accomplished by the $fH_2$-dependent reaction described by Eq. (5):

$$FeS_2 \text{ (pyrite)} + H_2 \rightleftharpoons FeS \text{ (pyrrothite)} + H_2S \qquad (5)$$

with $H_2$ being obviously available through the water dissociation according to Eq. (6):

$$2H_2O \rightleftharpoons 2H_2 + O_2 \qquad (6)$$

Thus, the initial sulfur-free aqueous fluid adjusts its $H_2S/H_2O$ fraction until the equilibrium with the $fH_2$ imposed by the buffer is reached. Likewise, the oxygen fugacity ($fO_2$) is constrained directly in the outer capsule by the used buffer and indirectly in the inner capsule because of the water dissociation. In the inner capsule, however, the $fO_2$ will be slightly lower compared to that imposed by the buffering assemblage in the outer capsule, as the fluid is not pure $H_2O$, with a consequent declined fugacity for $H_2O$ (and, consequently, for $O_2$).

Once filled, the capsules were welded shut in a frozen steel holder to prevent overheating and water loss. They were then reweighed to ensure that no fluid loss occurred during welding.

One synthesis (SOH-IW1) with a runtime of 6 h and two syntheses (SOH-IW2, SOH-FMQ1) with runtimes of 5 h were performed at 3 GPa and 700 °C, using an end-loaded piston cylinder apparatus. Details on the syntheses, including the experimental setup, are reported in Table 1. Temperatures were measured with K-type thermocouples and considered accurate to ±5 °C. Pressure calibration is based on the quartz/coesite transition (accuracy ±0.01 GPa)[17]. Samples were first brought to the run pressure (3 GPa), then heated to 700 C, with a ramp of 100 C/min. Experiments were quenched by turning off the power supply, resulting in a cooling rate of >40 °C/s. After quenching, the capsules were recovered; the outer capsule was peeled off, exposing the inner capsule, and dried in a vacuum oven at 110 °C for 1 h to remove the residual water trapped in the buffer.

The non-condensable volatiles, including $H_2S$ and $SO_2$ (Supplementary Figs. 5 and 6), and water in the inner capsule were quantitatively determined by the unique capsule-piercing quadrupole mass spectrometry (QMS) technique[6] (Supplementary Fig. 7) available at the University of Milan. Further details on this technique and on the sulfur volatiles measurement are provided in Supplementary Methods. At the investigated concentrations, the analytical uncertainty is <1 mol% for $H_2O$, while for $H_2S$, we estimated a LOD of 0.21 μmol and a LOQ of 0.64 μmol.

### Thermodynamic modeling of fluid composition

To validate our results, we compared the measured fluid composition with that predicted by thermodynamic modeling at the experimental conditions. Similar to the COH system, in the SOH system, the equilibrium composition of a fluid is fixed once the parameters $P$, $T$, and $fO_2$ (or $fH_2$) are known. However, contrary to what happens in COH fluids, which are saturated with graphite/diamond and therefore the activity of C is always equal to 1, SOH fluids are never in equilibrium with elemental S, because in the Earth's interior S tends to bond with siderophile elements, primarily iron, forming sulfides. Buffering for sulfur activity (or $fS_2$) can therefore only occur with the simultaneous presence of two different sulfides, such as FeS and $FeS_2$, a condition that was met in our experiments.

We started first to calculate oxygen and hydrogen fugacities in the outer capsule, containing the buffering IW + water or FMQ + water. $fO_2$ and $fH_2$ of redox-buffered water have been calculated at $P$–$T$ conditions of 3 GPa and 700 °C using the Perple_X package, considering the thermodynamic dataset of Holland and Powell revised by the authors in 2004 (hp04ver.dat) and an MRK equation of state for the H–O system. Then, the speciation of sulfide-saturated fluids has been calculated by fixing $fH_2$, which is homogeneous in the inner and in the outer capsule, and $fS_2$ constrained by FeS + $FeS_2$, using the Perple S–O–H equation of state of Connolly and Cesare.

Results of thermodynamic modeling for IW- and FMQ-buffered conditions are presented in Table 1.

## Data availability

The authors declare that the data supporting the findings of this study are available within the article.

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

## Acknowledgements

We thank Andrea Risplendente for assistance during the microprobe analysis and Stefano Piccin (University of Milan) for performing the image

analysis. Silvio Rosignoli helped in preparing some of the experiments. Dr. Teresa Schauperl is acknowledged for editorial handling. This paper benefited from the comments of three anonymous reviewers, who significantly improved the initial version of this manuscript. This study was supported by the Italian program MIUR PRIN 20224YR3AZ and the MIUR project "Dipartimenti di Eccellenza 2023-2027". The authors also acknowledge the support of the APC central fund of the university of Milan.

## Author contributions

A.S. wrote the manuscript, performed the experiments, and is responsible for data reduction and analysis. Q.M.S. analyses were conducted by S.T. and S.R. Thermodynamic modeling was performed by S.T. and L.T. The study was conceived by S.T.

## Competing interests

The authors declare no competing interests.
