## [Transparent Peer Review file · Communications Chemistry]

Analytical protocol for measuring micro-molar quantities of sulfur volatile species in experimental high pressure and temperature fluids

Corresponding Author: Dr Arianna Secchiari

Version 0:

Reviewer comments:

Reviewer #1

(Remarks to the Author)

Secchiari et al present a system to measure the sulfur volatile fluid recovered from piston-cylinder experiment. Measuring volatile in recovered samples are technically challenging. In my point of view, research in this topic is a welcome contribution to this forum for its wide implications in geochemistry. Their discussion on the experimental runtime, seal of H₂S, and chemical re-equilibrium may have provided very useful information for future S volatile characterization. It should be emphasized that such important implications need strong supporting data to convince potential readers. However, the current three single experiments, with major conclusion drawn from the first two (IW1 and IW2), are too thin to match its far-reaching implications. The two short paragraphs of conclusions contain little helpful information and definitely need overhaul. These major technical concerns below should be fully addressed before publication.

What are the major innovations in the design of current system, comparing to the one for COH? Sulfur's high reactivity is certainly a challenge. As I reader, I am more interested in the modification being made on the established COH system to address this challenge. In my point of view, both double-Au capsule and the capsule-piercing QMS technique have already been well reported.

For experiments with longer runtime (8-10 hours), the authors found capsule weakening and fluid leakage. This observation is very important but unfortunately, may soften the implication of the system. All data presented in this work was obtained at 3 GPa and 700 C, but capsule weakening has not been tested under other thermodynamic conditions. If this effect becomes prominent (I hope not) at a different pressure or temperature, the running conditions of this system will be limited.

The authors underscore the runtime in determining capsule resistance. Beyond the threshold, mentioned as 5 hours, H₂S begins to weaken the capsule. It is highly speculative with only two data points, to say 5 hours is the hard limit (line 210), as chemical equilibrium is modulated by pressure, temperature, runtime, sample heterogeneity, free energy barriers and many other factors.

I would like to see more details on the sample assemblage, particularly on the sealing of capsule. If the authors used welding, the generated heat may vaporize the water in between inner and outer capsules. The worst scenario is the loss of redox control and mistakenly induce the volatile composition difference between IW1 and IW2. That means even iron and wüstite were present in the buffer assemblage (Fig. 2ab), they still fail to effectively control the redox conditions. Sometimes Mg(OH)₂ was used instead of H₂O.

I am very interested in the volatile species from run SOH-IW2. Reaction (5) shows the chemical equilibrium of H₂S, H₂O, H₂ and SO₂. However, the cited reference of reaction (5) was performed under ambient pressure and > 200 C. Whether this reaction can occur under 3 GPa?

The authors claimed that these changes "were originated from late-stage chemical re-equilibration during quenching to room temperature and pressure". However, the samples were temperature quenched by cutting off power, then recovered to ambient pressure. During these procedures, the sample would be either at 3 GPa with rapid cooling temperature, or declining pressure (3 GPa to 1 atm) with room temperature. It is unlikely to meet the thermodynamic conditions described by ref 17 to implement reaction (5). I think the authors need to clarify the H₂/H₂O discrepancy better.

H₂ is more diffusive than H₂S in most situations. The only reason I could think of, is the gold capsule that seals H₂. But it is very surprising that the assembly has successfully sealed H₂, but not H₂S.

Table 1, what is bdl? "Below detection limit"?

Reviewer #2

(Remarks to the Author)

The authors conducted HP-HT experiment on sulfur volatile species to bridge knowledge gap between detection of products and their thermodynamics. The analytical protocol seems to be novel, but obtained results are insufficient to establish reliable thermodynamics of volatile species. Furthermore, the manuscript is not well organized, providing difficulties in reading and understanding for general readers. Thus, I do not recommend the publication in Communications Chemistry with current form.

- 1) I could not understand the overall procedure of HP-HT experiment. Did the authors perform mass spectral analyses during the HP-HT experiment? If it is right, the authors are encouraged to re-draw the schematics of procedure of HP-HT experiment.
- 2) The objective, obtained results, and impacts of this study are incomprehensible.
- 3) The solid phases shown in Figure 2 are helpful to establish thermodynamics. I would like to recommend Rietveld analyses for such phases.

Reviewer #3

(Remarks to the Author)

The authors present a novel methodology for measuring ex situ trace amounts of sulfur volatile species produced during experiments in high pressure-temperature conditions mimicking those present within the Earth. Volatile species and their presence and cycling within the Earth is one of the most ardent problems in contemporary Geo and planetary sciences, having direct implications for our understanding of both deep Earth processes (core composition, magnetic field generation and variation, etc.) and surface ones including climate regulation and its impact on human and non-human life. The presented method is a welcome addition to existing ones which are less quantitatively accurate or chemically specific.

The manuscript is generally well-written and clear. The authors do a very good job contextualising their work within the current state of the field, underscoring the lack of information on S-bearing systems despite their relevance, as well as highlighting the need for novel methods to address pressing issues and overcome existing limitations. The authors' method is likely to be of interest to wide audiences from both geology and high-pressure condensed matter physics and chemistry, and the results presented (as well as potentially future ones) are likely to be of widespread interest in Earth and planetary science, shedding light on fundamental solid-fluid systems.

The synthesis procedure is described in adequate detail to allow replication by a third party. The authors also describe the rationale and discovered caveats of commonly employed capsule materials which for the particular case of Sulfur would prove inadequate. The general method is described critically, and with sufficient detail and attention to physical and chemical considerations that may impact the obtained products or the validity of the measurements.

The results are presented in a critical and comparative fashion, the authors showing a systematic approach to isolate the effects of runtime from those of temperature and pressure. They very clearly re-emphasize the critical importance of controlled runtimes on the synthesised product and explain deviations from predictions through well-argued chemical reasoning. Comparisons with established thermodynamic models work both to strengthen the authors' arguments for the validity and accuracy of their method, and to showcase limitations of existing modelling approaches. This could be even further strengthened if a brief discussion of competing thermodynamic models is included (or a more robust justification for why the chosen ones were used).

The conclusions are well supported by the data and reasoning presented throughout the manuscript and supplementary information, however I feel these could be strengthened significantly by the authors not focussing here only on the methodology, but also re-emphasising and contextualising the actual scientific findings relating to Sulfur and the specific systems considered in the study, and the importance of chemical and physical variables on obtained products. In particular, more focus should be placed in the conclusion on agreement and also discrepancies between the experimental results presented and thermodynamic modelling.

On a more technical side, the authors mention that the 'quenching' is done by turning off the power supply, achieving a cooling rate of 40C/s. I think this could benefit from some more detail, and also a comment on the limitations of this approach. Specifically, is the cooling rate 40C/s on average, and if so, over how many seconds? Is the cooling rate actually time-dependent, with increasingly longer times to cool each subsequent degree as one approaches temperatures closer to room temperature? Can the authors provide perhaps a figure in the supplementary information showing the evolution of the measured sample temperature against time, both on heating and on cooling? Also, is the use of just switched off power source for cooling a practical/technical limitation of the experimental setup (no available means to induce faster rates/N2 cooling), or a desired behaviour due to the particular geological samples considered here having specific kinetics? I strongly believe addressing these would strengthen the argument for the proposed methodology even further, as the presented technique could be of wide interest outside geology, for example in high-pressure physics/chemistry, where samples might

actually need much faster quenching rates to be arrested in the relevant state.

Overall, I think this is a very good manuscript for publication in your journal, which could be improved if the (minor) points raised above have been addressed.

Version 1:

Reviewer comments:

Reviewer #1

(Remarks to the Author)

The authors have done a good job in answering my questions. The manuscript can be published.

Reviewer #2

(Remarks to the Author)

The authors properly revised the manuscript, thereby I would like to recommend the publication with the current form.

Reviewer #3

(Remarks to the Author)

I am satisfied with the changes made by the authors to their manuscript in response to the previous comments, and I believe it is now in a suitable state for publication.

Reviewers' Comments:

Reviewer #1 (Remarks to the Author):

Secchiari et al present a system to measure the sulfur volatile fluid recovered from piston-cylinder experiment. Measuring volatile in recovered samples are technically challenging. In my point of view, research in this topic is a welcome contribution to this forum for its wide implications in geochemistry. Their discussion on the experimental runtime, seal of H₂S, and chemical re-equilibrium may have provided very useful information for future S volatile characterization. It should be emphasized that such important implications need strong supporting data to convince potential readers. However, the current three single experiments, with major conclusion drawn from the first two (IW1 and IW2), are too thin to match its far-reaching implications. The two short paragraphs of conclusions contain little helpful information and definitely need overhaul. These major technical concerns below should be fully addressed before publication.

What are the major innovations in the design of current system, comparing to the one for COH? Sulfur's high reactivity is certainly a challenge. As I reader, I am more interested in the modification being made on the established COH system to address this challenge. In my point of view, both double-Au capsule and the capsule-piercing QMS technique have already been well reported.

For experiments with longer runtime (8-10 hours), the authors found capsule weakening and fluid leakage. This observation is very important but unfortunately, may soften the implication of the system. All data presented in this work was obtained at 3 GPa and 700 C, but capsule weakening has not been tested under other thermodynamic conditions. If this effect becomes prominent (I hope not) at a different pressure or temperature, the running conditions of this system will be limited.

The authors underscore the runtime in determining capsule resistance. Beyond the threshold, mentioned as 5 hours, H₂S begins to weaken the capsule. It is highly speculative with only two data points, to say 5 hours is the hard limit (line 210), as chemical equilibrium is modulated by pressure, temperature, runtime, sample heterogeneity, free energy barriers and many other factors.

I would like to see more details on the sample assemblage, particularly on the sealing of capsule. If the authors used welding, the generated heat may vaporize the water in between inner and outer capsules. The worst scenario is the loss of redox control and mistakenly induce the volatile composition difference between IW1 and IW2. That means even iron and wüstite were present in the buffer assemblage (Fig. 2ab), they still fail to effectively control the redox conditions. Sometimes Mg(OH)₂ was used instead of H₂O.

I am very interested in the volatile species from run SOH-IW2. Reaction (5) shows the chemical equilibrium of H₂S, H₂O, H₂ and SO₂. However, the cited reference of reaction (5) was performed under ambient pressure and > 200 C. Whether this reaction can occur under 3 GPa?

The authors claimed that these changes "were originated from late-stage chemical re-equilibration during quenching to room temperature and pressure". However, the samples were temperature quenched by cutting off power, then recovered to ambient pressure. During these procedures, the sample would be either at 3 GPa with rapid cooling temperature, or declining pressure (3 GPa to 1 atm) with room temperature. It is unlikely to meet the thermodynamic conditions described by ref 17 to implement reaction (5). I think the authors need to clarify the H₂/H₂O discrepancy better.

H₂ is more diffusive than H₂S in most situations. The only reason I could think of, is the gold capsule that seals H₂. But it is very surprising that the assembly has successfully sealed H₂, but not H₂S.

Table 1, what is bdl? "Below detection limit"?

Reviewer #2 (Remarks to the Author):

The authors conducted HP-HT experiment on sulfur volatile species to bridge knowledge gap between detection of products and their thermodynamics. The analytical protocol seems to be novel, but obtained results are insufficient to establish reliable thermodynamics of volatile species. Furthermore, the manuscript is not well organized, providing difficulties in reading and understanding for general readers. Thus, I do not recommend the publication in Communications Chemistry with current form.

- 1) I could not understand the overall procedure of HP-HT experiment. Did the authors perform mass spectral analyses during the HP-HT experiment? If it is right, the authors are encouraged to re-draw the schematics of procedure of HP-HT experiment.
- 2) The objective, obtained results, and impacts of this study are incomprehensible.
- 3) The solid phases shown in Figure 2 are helpful to establish thermodynamics. I would like to recommend Rietveld analyses for such phases.

Reviewer #3 (Remarks to the Author):

The authors present a novel methodology for measuring ex situ trace amounts of sulfur volatile species produced during experiments in high pressure-temperature conditions mimicking those present within the Earth. Volatile species and their presence and cycling within the Earth is one of the most ardent problems in contemporary Geo and planetary sciences, having direct implications for our understanding of both deep Earth processes (core composition, magnetic field generation and variation, etc.) and surface ones including climate regulation and its impact on human and non-human life. The presented method is a welcome addition to existing ones which are less quantitatively accurate or chemically specific.

The manuscript is generally well-written and clear. The authors do a very good job contextualising their work within the current state of the field, underscoring the lack of information on S-bearing systems despite their relevance, as well as highlighting the need for novel methods to addressing pressing issues and overcome existing limitations. The authors' method is likely to be of interest to wide audiences from both geology and high-pressure condensed matter physics and chemistry, and the results presented (as well as potentially future ones) are likely to be of widespread interest in Earth and planetary science, shedding light on fundamental solid-fluid systems.

The synthesis procedure is described in adequate detail to allow replication by a third party. The authors also describe the rationale and discovered caveats of commonly employed capsule materials which for the particular case of Sulfur would prove inadequate. The general method is described critically, and with sufficient detail and attention to physical and chemical considerations that may impact the obtained products or the validity of the measurements.

The results are presented in a critical and comparative fashion, the authors showing a systematic approach to isolate the effects of runtime from those of temperature and pressure. They very clearly re-emphasize the critical importance of controlled runtimes on the synthesised product and explain deviations from predictions through well-argued chemical reasoning. Comparisons with established thermodynamic models work both to strengthen the authors' arguments for the validity and accuracy of their method, and to showcase limitations of existing modelling approaches. This could be even further strengthened if a brief discussion of competing thermodynamic models is included (or a more robust justification for why the chosen ones were used).

The conclusions are well supported by the data and reasoning presented throughout the manuscript and supplementary information, however I feel these could be strengthened significantly by the authors not focussing here only on the methodology, but also re-emphasising and contextualising the actual scientific findings relating to Sulfur and the specific systems considered in the study, and the importance of chemical and physical variables on obtained products. In particular, more focus should be placed in the conclusion on agreement and also discrepancies between the experimental results presented and thermodynamic modelling.

On a more technical side, the authors mention that the 'quenching' is done by turning off the power supply, achieving a cooling rate of 40C/s. I think this could benefit from some more detail, and also a comment on the limitations of this approach. Specifically, is the cooling rate 40C/s on average, and if so, over how many seconds? Is the cooling rate actually time-dependent, with increasingly longer times to cool each subsequent degree as one approaches temperatures closer to room temperature? Can the authors provide perhaps a figure in the supplementary information showing the evolution of the measured sample temperature against time, both on heating and on cooling? Also, is the use of just switched off power source for cooling a practical/technical limitation of the experimental setup (no available means to induce faster rates/N₂ cooling), or a desired behaviour due to the particular geological samples considered here having specific kinetics? I strongly believe addressing these would strengthen the argument for the proposed methodology even further, as the presented technique could be of wide interest outside geology, for example is high-pressure physics/chemistry, where samples might actually need much faster quenching rates to be arrested in the relevant state.

Overall, I think this is a very good manuscript for publication in your journal, which could be improved if the (minor) points raised above have been addressed.

We sincerely appreciate the thoughtful comments from all three reviewers, who have provided valuable suggestions to improve this revised version of the manuscript. Below, we have listed detailed responses to each of the reviewers' comments.

Although the criticism raised by Reviewer #2, we appreciated that he/she acknowledges some of the major points of the manuscript: the realization “of experiments on sulfur volatile species to bridge knowledge gap between detection of products and their thermodynamics” and the novelty of the developed protocol.

To address the concerns raised by the three reviewers we have made significant improvements to the original manuscript.

To this purpose, we conducted **additional experiments to evaluate the potential effects of pressure and temperature on the capsule weakening effect** described throughout the paper. These experiments confirmed that while our protocol is optimized for P-T conditions of 3 GPa and 700°C, it can also be extended to other P-T ranges.

Moreover, in response to Reviewer #2's comments, we performed X-ray **image analysis** (Fig. 3, Supplementary Table 2) to better characterize the solid run products.

Overall, we believe that this revised version of the manuscript has been significantly improved compared to initial submission. With the proposed changes, including the addition of **new figures and tables** in the supplementary material, we are confident that we can demonstrate the robustness of the presented data and developed protocol to the reader.

Reviewer #1 (Remarks to the Author):

Secchiari et al present a system to measure the sulfur volatile fluid recovered from piston-cylinder experiment. Measuring volatile in recovered samples are technically challenging. In my point of view, research in this topic is a welcome contribution to this forum for its wide implications in geochemistry. Their discussion on the experimental runtime, seal of H₂S, and chemical re-equilibrium may have provided very useful information for future S volatile characterization. It should be emphasized that such important implications need strong supporting data to convince potential readers. However, the current three single experiments, with major conclusion drawn from the first two (IW1 and IW2), are too thin to match its far-reaching implications. The two short paragraphs of conclusions contain little helpful information and definitely need overhaul. These major technical concerns below should be fully addressed before publication.

In the revised version of the manuscript, we performed three additional experiments to better constrain the role of P-T conditions on the H₂S-forming reaction and the capsule weakening. Since we believe that these experiments constitute an important additional part of the manuscript, but not its main focus, we have decided to report them as Supplementary Material. Please refer to Supplementary Table 3 and the Supplementary Material for further detail.

While our work focuses on the results of three experiments, a total of 15 runs — plus a permeability test — were conducted (see Supplementary Table 1). These experiments, with durations ranging from one week to five hours, showed that equilibrium is achieved within 5 hours. For all IW-buffered experiments exceeding 5 hours, the capsule weakening effect, as described and discussed in the paper, was observed. This effect was evidenced by the absence of fluid in the capsule and by the characteristic smell of rotten eggs from the outer capsule. This demonstrates that H₂S is formed and progressively weakens the inner capsule after 5 hours, likely via diffusion. Ultimately, the H₂S is lost during quenching. This is demonstrated by microprobe analysis, which did not evidence any reaction between the produced sulfur-bearing volatile species and either the inner or outer capsule.

1) What are the major innovations in the design of current system, comparing to the one for COH? Sulfur's high reactivity is certainly a challenge. As I reader, I am more interested in the modification

being made on the established COH system to address this challenge. In my point of view, both double-Au capsule and the capsule-piercing QMS technique have already been well reported.

As the reviewer correctly pointed out, neither the use of QMS or the double Au-Au capsule is the novel aspect of our work.

Nevertheless, we would like to emphasize that although the double Au capsule setup has been used in experiments related to sulfur in magmatic systems, no one has ever tested its permeability to hydrogen under low temperatures (i.e. 700 °C). Through our preliminary permeability test (experiment SOH-FMQ0, see Supplementary Material), we experimentally demonstrated for the first time that the double Au capsule is permeable to hydrogen, representing an ideal setup for our protocol.

However, the true innovation of our study lies in the development of the first analytical protocol for accurate and precise measurements of sulfur volatile species experimentally produced under HT and HP conditions in small-volume (< 20 mm³) capsules. Despite the scientific relevance of this topic, no established protocols currently exist for ex-situ measurements of these volatile species in experimental fluids, likely due to the high reactivity of S, which makes such analyses particularly challenging.

To perform these analyses, we use the same analytical instrumentation previously described by Tiraboschi et al. (2016) and routinely employed for COH fluid measurements (Tumiati et al., 2017; 2020; 2022). The only modification to the QMS, as outlined in the supplementary material, involved the use of two gas mixtures for the standardization of the sulfur-bearing volatile species:

- 1) Ar + H₂S (1 vol.%)*
- 2) Ar + SO₂ (1 vol.%)*

Unlike the gas mixtures used for carbon species (Tiraboschi et al., 2016), which typically contain the species of interest at 10% concentration, sulfur-bearing volatile species in the gas mixtures are present at lower concentrations. This adjustment was necessary due to the aggressive nature of sulfur volatiles, which, if present in higher concentrations, could potentially damage the main spectrometer line.

2) For experiments with longer runtime (8-10 hours), the authors found capsule weakening and fluid leakage. This observation is very important but unfortunately, may soften the implication of the system. All data presented in this work was obtained at 3 GPa and 700 C, but capsule weakening has not been tested under other thermodynamic conditions. If this effect becomes prominent (I hope not) at a different pressure or temperature, the running conditions of this system will be limited.

To address this potential issue, we conducted a series of new experiments (see Supplementary Table 3 and Supplementary Material for further details) to evaluate the possible effects of T and P on capsule weakening.

In the first set of experiments, we assessed the effect of temperature, while keeping pressure constant. Temperature was set at 800°C (SOH-IW3) and 600°C (SOH-IW4). This range of temperatures was chosen as representative of the thermal gradients occurring in subduction zones (see Syracuse et al., 2010). Additionally, as the protocol was specifically designed for investigating fluid-solid equilibria, we maintained the temperature confidently below the melting temperature of the sulfide phase (T solidus for pyrrhotite ~ 900°C, Sharp et al., 1969 J. Geoph. Res.)

These new experiments revealed that kinetics is a key factor in governing the H₂S forming reaction. In the SOH-IW4 run, we measured a fluid composed entirely of H₂O, while in the SOH-IW3 run, the capsule was again found empty. This suggests that at 600°C kinetics is unfavorable, meaning the reaction occurs slowly — resulting in a final fluid with unchanged composition compared to the initial one. At this temperature, longer run durations would be required to achieve chemical equilibrium.

By contrast, at 800°C, the H₂S-forming reaction is kinetically favored, proceeding at faster rates compared to runs performed at 700°C. This leads to the capsule weakening effect within 5 hours, invariably attested by the typical rotten eggs smell and the empty capsule.

By contrast, in the run conducted at lower pressure (SOH-IW5) we analyzed a fluid composed of H₂O, H₂S, SO₂ and H₂. This result shows that pressure does not have a significant effect on capsule weakening, and the produced H₂S is retained within the inner capsule. Overall, the application of our protocol under different P-T conditions is feasible but requires further optimization of the run time, if the experiments are run at different temperatures. This optimization is however beyond the scope of this work and will be further explored in next works.

3) The authors underscore the runtime in determining capsule resistance. Beyond the threshold, mentioned as 5 hours, H₂S begins to weaken the capsule. It is highly speculative with only two data points, to say 5 hours is the hard limit (line 210), as chemical equilibrium is modulated by pressure, temperature, runtime, sample heterogeneity, free energy barriers and many other factors.

The 5-hour threshold was experimentally determined through 13 (IW-buffered) runs with varying durations, ranging from one week to 5 hours. We have added a new table in the Electronic Supplementary Material (Table S1) that includes all the experiments conducted, along with their run times. This specific 5-hour threshold applies to the P-T conditions of 3 GPa and 700°C, for which the protocol is designed and optimized.

4) I would like to see more details on the sample assemblage, particularly on the sealing of capsule. If the authors used welding, the generated heat may vaporize the water in between inner and outer capsules. The worst scenario is the loss of redox control and mistakenly induce the volatile composition difference between IW1 and IW2. That means even iron and wüstite were present in the buffer assemblage (Fig. 2ab), they still fail to effectively control the redox conditions. Sometimes Mg(OH)₂ was used instead of H₂O.

The preparation of both the IC and OC followed a specially designed procedure that has been thoroughly tested for COH fluids (Tumiati et al., 2017). After filling, the capsules were frozen and then welded shut while placed in a frozen steel holder to prevent overheating. To ensure no fluid loss during the welding process, all the capsules were weighed before and after welding. We have added a few sentences to explain the procedure in the Methods section (lines 127-128)

By contrast, using Mg(OH)₂ instead of water is not feasible, as it would introduce an additional component into the system, which must consist solely of sulfides and water. In any case, the presence of free water in the inner capsule after the run was measured and not merely assumed, while the presence of water in the outer capsule was always checked qualitatively by observing bubbling when the capsule was peeled.

5) I am very interested in the volatile species from run SOH-IW2. Reaction (5) shows the chemical equilibrium of H₂S, H₂O, H₂ and SO₂. However, the cited reference of reaction (5) was performed under ambient pressure and > 200 C. Whether this reaction can occur under 3 GPa? The authors claimed that these changes “were originated from late-stage chemical re-equilibration during quenching to room temperature and pressure”. However, the samples were temperature quenched by cutting off power, then recovered to ambient pressure. During these procedures, the sample would be either at 3 GPa with rapid cooling temperature, or declining pressure (3 GPa to 1 atm) with room temperature. It is unlikely to meet the thermodynamic conditions described by ref 17 to implement reaction (5). I think the authors need to clarify the H₂/H₂O discrepancy better.

Reaction (5) describes the formation of SO₂ from H₂S. As correctly pointed out by the reviewer, this reaction occurs at hydrothermal conditions (Ma et al., 2010). As the bulk fluid composition was preserved in experiment SOH-IW2, the conditions suitable for the reaction (5) are likely achieved during the final stages of the experiment, i.e. during quenching. It has been demonstrated that ex-situ experiments preserve the bulk composition of the fluid; however, the fluid speciation, which depends on P-T conditions, may not be maintained during quenching. Therefore, ex-situ experiments can only determine the final bulk composition of the

fluid. Rapid quench techniques, using faster cooling rates, represent the frontier of experimental petrology, allowing in the next future to overcome these possible issues related to cooling during quenching. To this purpose, our laboratory will be soon implemented with a piston-cylinder apparatus equipped with a rapid quench, which will be for sure of great interest for the future development of SOH fluids investigation. We have added a few sentences in the manuscript: lines 222-227.

6) H₂ is more diffusive than H₂S in most situations. The only reason I could think of, is the gold capsule that seals H₂. But it is very surprising that the assembly has successfully sealed H₂, but not H₂S.

H₂ is more easily diffusible, as demonstrated by run SOH-IW1, where a portion of the produced H₂S remains in the capsule, while H₂ diffused completely. In run SOH-IW2, where the fluid achieved thermodynamic equilibrium and the capsule was not weakened, H₂ was present and analyzed in the final fluid.

Table 1, what is bdl? "Below detection limit"?

Yes, we have added the acronym to the table caption.

Reviewer #2 (Remarks to the Author):

The authors conducted HP-HT experiment on sulfur volatile species to bridge knowledge gap between detection of products and their thermodynamics. The analytical protocol seems to be novel, but obtained results are insufficient to establish reliable thermodynamics of volatile species. Furthermore, the manuscript is not well organized, providing difficulties in reading and understanding for general readers. Thus, I do not recommend the publication in Communications Chemistry with current form.

Our work aims to develop a new protocol for measuring the sulfur volatile species experimentally produced under HP-HT conditions. In contrast, building a new thermodynamic model is beyond the focus of this study. However, to develop a reliable and robust protocol, validate our results against the existing thermodynamic models is essential. This was done by comparing the measured bulk fluid compositions with compositions predicted by thermodynamics.

1) I could not understand the overall procedure of HP-HT experiment. Did the authors perform mass spectral analyses during the HP-HT experiment? If it is right, the authors are encouraged to re-draw the schematics of procedure of HP-HT experiment.

In this work, we conducted syntheses under HP-HT conditions in low-volume experimental capsules using a piston cylinder apparatus. Every synthesis was followed by quantitative analysis of the synthesized fluids using a QMS equipped with a capsule-piercing device. The characterization of the solid run products was performed using an electron microscope. The procedure was illustrated in the Method section and in the Electronic Supplementary Material.

2) The objective, obtained results, and impacts of this study are incomprehensible.

As geologists, we are interested in investigating processes occurring deep within the Earth's interior that are not directly accessible to our observation. While our protocol may be used for materials science, it has been specifically designed to investigate fluid-solid phase interactions and equilibria in natural environments (subduction zones) under HP-HT conditions (3 GPa, 700 °C). This research has significant implications for the geochemical cycling of volatile elements within the deep Earth interior, particularly regarding Earth's climate regulation.

3) The solid phases shown in Figure 2 are helpful to establish thermodynamics. I would like to recommend Rietveld analyses for such phases.

Although it is not possible to perform routine Rietveld analysis because the synthesized material is available in the order of a few milligrams with an inhomogeneous distribution in the capsule, we appreciate the reviewer's comment and suggestion. This inspired us to conduct image analysis on solid run products based on X-rays maps obtained using electron microprobe. Maps are shown in Fig. 3. Results of the image analysis are reported in Supplementary Table 2 and discussed in the text.

Reviewer #3 (Remarks to the Author):

The authors present a novel methodology for measuring ex situ trace amounts of sulfur volatile species produced during experiments in high pressure-temperature conditions mimicking those present within the Earth. Volatile species and their presence and cycling within the Earth is one of the most ardent problems in contemporary Geo and planetary sciences, having direct implications for our understanding of both deep Earth processes (core composition, magnetic field generation and variation, etc.) and surface ones including climate regulation and its impact on human and non-human life. The presented method is a welcome addition to existing ones which are less quantitatively accurate or chemically specific.

The manuscript is generally well-written and clear. The authors do a very good job contextualising their work within the current state of the field, underscoring the lack of information on S-bearing systems despite their relevance, as well as highlighting the need for novel methods to addressing pressing issues and overcome existing limitations. The authors' method is likely to be of interest to wide audiences from both geology and high-pressure condensed matter physics and chemistry, and the results presented (as well as potentially future ones) are likely to be of widespread interest in Earth and planetary science, shedding light on fundamental solid-fluid systems.

The synthesis procedure is described in adequate detail to allow replication by a third party. The authors also describe the rationale and discovered caveats of commonly employed capsule materials which for the particular case of Sulfur would prove inadequate. The general method is described critically, and with sufficient detail and attention to physical and chemical considerations that may impact the obtained products or the validity of the measurements.

1) The results are presented in a critical and comparative fashion, the authors showing a systematic approach to isolate the effects of runtime from those of temperature and pressure. They very clearly re-emphasize the critical importance of controlled runtimes on the synthesised product and explain deviations from predictions through well-argued chemical reasoning. Comparisons with established thermodynamic models work both to strengthen the authors' arguments for the validity and accuracy of their method, and to showcase limitations of existing modelling approaches. This could be even further strengthened if a brief discussion of competing thermodynamic models is included (or a more robust justification for why the chosen ones were used).

To the best of our knowledge, no alternative thermodynamic models are available for predicting the composition and speciation of S–O–H fluids other than the one used in this study.

2) The conclusions are well supported by the data and reasoning presented throughout the manuscript and supplementary information, however I feel these could be strengthened significantly by the authors not focussing here only on the methodology, but also re-emphasising and contextualising the actual scientific findings relating to Sulfur and the specific systems considered in the study, and the importance of chemical and physical variables on obtained products. In particular, more focus should be placed in the conclusion on agreement and also discrepancies between the experimental results presented and thermodynamic modelling.

We have modified and implemented the Conclusions section following the reviewer's suggestion.

3) On a more technical side, the authors mention that the 'quenching' is done by turning off the power supply, achieving a cooling rate of 40C/s. I think this could benefit from some more detail, and also a comment on the limitations of this approach. Specifically, is the cooling rate 40C/s on average, and if so, over how many seconds? Is the cooling rate actually time-dependent, with increasingly longer times to cool each subsequent degree as one approaches temperatures closer to room temperature? Can the authors provide perhaps a figure in the supplementary information showing the evolution of the measured sample temperature against time, both on heating and on cooling? Also, is the use of just switched off power source for cooling a practical/technical limitation of the experimental setup (no available means to induce faster rates/N₂ cooling), or a desired behaviour due to the particular geological samples considered here having specific kinetics? I strongly believe addressing these would strengthen the argument for the proposed methodology even further, as the presented technique could be of wide interest outside geology, for example in high-pressure physics/chemistry, where samples might actually need much faster quenching rates to be arrested in the relevant state.

During quenching, cooling occurs at a linear rate of 40°C/s until the temperature reaches 50°C. This technique is routinely used in experimental petrology laboratories and has been proven effective in preserving the bulk fluid composition for COH fluids, as extensively demonstrated in the scientific literature (Tumiati et al., 2017, 2020, 2022; Peng et al., 2022).

Rapid quench techniques, using faster cooling rates, represents a frontier in experimental petrology (Bondar et al., 2022). Although promising, these techniques are in their early stages of implementation. We have added a few sentences regarding these techniques in the revised version of the manuscript (lines 222-227).

The piston-cylinder apparatus available at Milan University will be soon upgraded with a rapid quench system, which will be of great interest for the study of SOH fluids.

Overall, I think this is a very good manuscript for publication in your journal, which could be improved if the (minor) points raised above have been addressed.

We hope that with the improvements introduced in this revised version of the manuscript, the reviewer will be convinced of the robustness and wider applicability of our designed protocol.